# Carnosol, a Rosemary Ingredient Discovered in a Screen for Inhibitors of SARM1-NAD^+^ Cleavage Activity, Ameliorates Symptoms of Peripheral Neuropathy

**DOI:** 10.3390/antiox14070808

**Published:** 2025-06-30

**Authors:** Hitoshi Murata, Kazuki Ogawa, Yu Yasui, Toshiki Ochi, Nahoko Tomonobu, Ken-Ichi Yamamoto, Rie Kinoshita, Yoji Wada, Hiromichi Nakamura, Masahiro Nishibori, Masakiyo Sakaguchi

**Affiliations:** 1Department of Cell Biology, Okayama University Graduate School of Medicine, Dentistry and Pharmaceutical Sciences, 2-5-1 Shikata-cho, Kita-ku, Okayama 700-8558, Japanmasa-s@md.okayama-u.ac.jp (M.S.); 2Tama Biochemical Co., Ltd., 1-23-3 Nishi-shinjuku, Shinjyuku-ku, Tokyo 160-0023, Japan; 3Department of Translational Research and Drug Development, Okayama University Graduate School of Medicine, Dentistry and Pharmaceutical Sciences, 2-5-1 Shikata-cho, Kita-ku, Okayama 700-8558, Japan

**Keywords:** SARM1, carnosol, NAD^+^, axon degeneration, peripheral neuropathy

## Abstract

Sterile alpha and Toll/interleukin receptor motif-containing protein 1 (SARM1) is a nicotinamide adenine dinucleotide (NAD^+^) hydrolase involved in axonal degeneration and neuronal cell death. SARM1 plays a pivotal role in triggering the neurodegenerative processes that underlie peripheral neuropathies, traumatic brain injury, and neurodegenerative diseases. Importantly, SARM1 knockdown or knockout prevents the degeneration; as a result, SARM1 has been attracting attention as a potent therapeutic target. In recent years, the development of several SARM1 inhibitors derived from synthetic chemical compounds has been reported; however, no dietary ingredients with SARM1 inhibitory activity have been identified. Therefore, we here focused on dietary ingredients and found that carnosol, an antioxidant contained in rosemary, inhibits the NAD^+^-cleavage activity of SARM1. Purified carnosol inhibited the enzymatic activity of SARM1 and suppressed neurite degeneration and cell death induced by the anti-cancer medicine vincristine (VCR). Carnosol also inhibited VCR-induced hyperalgesia symptoms, suppressed the loss of intra-epidermal nerve fibers in vivo, and reduced the blood fluid level of phosphorylated neurofilament-H caused by an axonal degeneration event. These results indicate that carnosol has a neuroprotective effect via SARM1 inhibition in addition to its previously known antioxidant effect via NF-E2-related factor 2 and thus suppresses neurotoxin-induced peripheral neuropathy.

## 1. Introduction

Axon degeneration is an early hallmark of multiple neurological diseases, including neuropathies, traumatic brain injury, and neurodegenerative diseases such as Alzheimer’s disease, Parkinson’s disease, and amyotrophic lateral sclerosis (ALS) [1,2,3]. Inhibition of axon degeneration may lead to the inhibition of disease progression and treatment; thus, research is being conducted into the inhibition of axon degeneration as a drug discovery target. In recent years, the mechanism underlying Wallerian degeneration, an injury-induced process of anterograde axon degeneration, has been elucidated, yielding promising targets for axon protection [1]. Nerve transection and nerve crush models have traditionally been used to study Wallerian degeneration [4]; however, chemical insults have been shown to provoke a degeneration akin to Wallerian degeneration [5,6,7]. Axon degeneration in neurodegenerative disease follows the same pathway; the axons progressively degenerate in a “dying back” process [8,9].

Wallerian degeneration involves an active program of disassembly of the injured axon. Genetic screens identified sterile alpha and toll/interleukin (TIR) motif-containing protein 1 (SARM1) as a prodegenerative factor [10]. SARM1 is the central driver of an evolutionarily conserved program of axonal degeneration downstream of chemical, inflammatory, mechanical, or metabolic insults to the axon [11,12,13]. From the N-terminus, SARM1 is composed of a mitochondrial transport signal, an auto-inhibitory ARM domain, tandem repeat SAM domains that induce oligomerization of the protein, and a TIR domain that hydrolyzes NAD^+^ to nicotinamide (NAM), ADPR, and cyclic ADPR [14,15,16]. SARM1 is activated in response to external stimuli, depleting NAD^+^ and causing axonal degeneration and cell death [17,18,19]. The genetic elimination of SARM1 significantly reduces axonal degeneration, attenuates traumatic brain injury, and mitigates axonal loss induced by chemotherapeutic agents, such as vincristine (VCR), paclitaxel and oxaliplatin, or diabetic metabolic conditions [20,21,22,23,24,25,26]. The deletion of SARM1 has also been reported to alleviate symptoms of Alzheimer’s disease, Parkinson’s disease, and ALS, suggesting that it may be effective in treating neurodegenerative diseases [27,28,29]. Therefore, drug discovery research targeting SARM1 is being actively conducted. Several SARM1 inhibitors have been reported so far. Some inhibitors act as pseudo substrates and are converted by SARM1 into NAD^+^ mimetics, which become active inhibitors and inhibit the NAD^+^-cleavage activity of the SARM1-TIR domain [30,31]. Inhibitors targeting the ARM domain, an autoinhibitory domain, have also been reported; they have been shown to keep the ARM domain in an inactive form by binding to cysteine residues or forming octamers [32,33,34].

We hypothesized that there may still be unidentified compounds with SARM1 inhibitory effects in dietary ingredients that can be consumed on a daily basis. To investigate this, we screened various candidate compounds for their SARM1-inhibition potential, resulting in the identification of one potential candidate, rosemary, due to its highest competency in blocking SARM1 activity among those tested. Rosemary, a plant of the Lamiaceae family and an herb used in cooking and flavoring, has long been known to have neuroprotective effects due to its antioxidant and anti-inflammatory properties. Rosemary has been reported for its unusual druggable role in neuronal protection via multiple mechanisms. For instance, it suppresses the amyloid beta-induced decline in mitochondrial membrane potential and reduces rotenone-induced lipid peroxidation and oxidative stress [35,36]. Attributing to the medicinal-gifts trait of the herb as mentioned above, it has been studied for its potential in treating some neurodegenerative diseases, culminating in showing beneficial effects on Alzheimer’s disease and Parkinson’s disease [37]. What is the medicinal nature of rosemary in these intractable diseases? The neuroprotective components of rosemary that have been reported are as follows: rosmarinic acid, ursolic acid, carnosic acid, and carnosol. Rosmarinic acid neutralizes free radicals and protects neurons from oxidative stress. It has been shown to play roles in the suppression of inflammatory markers and promotion of BDNF production [38,39]. In addition, rosmarinic acid has been shown to exhibit neuroprotective effects on glutamatergic synapses by regulating oxidative phosphorylation in MPTP-induced Parkinson’s disease model mice, as demonstrated by a proteomic approach in the substantia nigra of the disease model [40]. Ursolic acid, which suppresses the production of inflammatory cytokines and microglia activation, presents another exciting prospect [41]. As for carnosic acid and carnosol, they are primarily known for neuroprotective effects mediated by the antioxidant transcription factor, NF-E2-related factor 2 (NRF2) [42,43]. NRF2 normally binds to Kelch-like ECH-associated protein 1 (KEAP1) and its expression level is kept low; carnosic acid and carnosol inhibit KEAP1 and activate NRF2 [44,45]. Activation of NRF2 up-regulates gene expression of antioxidant and detoxification enzymes, protecting neurons from oxidative stress and inflammation. Carnosol has also been shown to improve protein homeostasis and mitochondrial homeostasis, and to reduce pathogenic protein aggregation and cognitive impairment [46]. In sum, rosemary has many neuroprotective components with the potential to inhibit SARM1 activity.

We herein report for the first time that carnosol, an antioxidant component of rosemary, specifically inhibits the NAD^+^-cleavage activity of SARM1. Through its inhibition of NAD^+^-cleavage activity, carnosol inhibits SARM1-dependent cell death and axonal degeneration, reducing vincristine (VCR)-induced pain and peripheral neuropathy symptoms in vivo. In our in vivo experiments, we observed a significant reduction in pain and neuropathy symptoms in the test subjects treated with carnosol, compared to the control group. This suggests that carnosol, an ingredient of rosemary, could be a potential therapeutic agent for chemotherapy-induced peripheral neuropathy (CIPN) and neurodegenerative diseases.

## 2. Materials and Methods

### 2.1. Chemicals and Antibodies

Rosmarinic acid, ursolic acid, and vincristine sulfate were purchased from FUJIFILM Wako Pure Chemical Corporation (Osaka, Japan). Vacor (Pyrinuron) was purchased from Chem Service (West Chester, PA, USA). The following antibodies were used: rabbit anti-SARM1 (Cell Signaling Technology, Beverly, MA, USA [CST]; cat#13022), rabbit anti-AMPKα (CST; cat#5832), rabbit anti-phospho-AMPKα (Thr172) (CST; cat#2535), mouse anti-Neurofilament-M ([NF-M] CST; cat#2838), rabbit anti-cleaved caspase-3 (Cl. Caspase3) (CST; cat#9664), rabbit anti-β3 tubulin (CST; cat#5568), mouse anti-NMNAT2 (Santa Cruz Biotechnology, Santa Cruz, CA, USA; cat#sc-515206), mouse anti-neurofilament-L (NF-L) (Santa Cruz Biotechnology, cat#sc-20012), rabbit anti-protein gene product 9.5 (PGP9.5) (Proteintech, Rosemont, IL, USA; cat#14730-1-AP), rabbit anti-FLAG (DDDDK-tag) (MBL, Nagoya, Japan; cat#PM020), and mouse anti-β-actin (Sigma-Aldrich, Tokyo, Japan; cat#A2228). The mouse monoclonal antibody (mAb) against phospho-Ser548 SARM1 was generated by ITM Co. (Matsumoto, Japan).

### 2.2. Extraction of Rosemary Ingredients

A total of 3000 g of raw rosemary materials were added to 24 L of 80% ethanol (EtOH) in water (8 times the volume of the charged material); circulation extraction was performed at 70 °C for 1.5 h and the contents were extracted and concentrated. Next, 18 L of 80% EtOH in water (6 times the volume of the charged material) was added; circulation extraction was performed in the same manner and the contents were extracted and concentrated. This operation was repeated twice, for a total of four extractions with 80% EtOH in water. Then, the full amount of the extract was concentrated; the EtOH was distilled off to obtain a crude concentrate containing 991.59 g of rosemary extract solids. A total of 10 L of water was added to the crude concentrate (solid content 991.59 g), and the mixture was stirred in a 60 °C-water bath for 1 h and then filtered (SUS Nutsche φ360 mm, No. 2 filter paper, precoated with 500 g of diatomaceous earth (Radiolite #900)). Cake (1) on the filter paper was washed with 2.5 L of water and the filtrate and washings were combined to obtain the “water extract” (aqueous solution containing 511.39 g of solid content). This water extract was directly passed through 5 L of SP825L resin (solid content load 10%). Then, 20 L of water (water washing fraction), 20 L of 40% EtOH in water (desorption fraction), and 15 L of 70% EtOH in water (washing fraction) were passed through in that order. The 40% EtOH in water (desorption fraction) was dried as the main fraction and RME–RA (rosemary extract–rosmarinic acid) was obtained.

Next, cake (1) was collected, 10 L of 70% EtOH in water was added and the mixture was stirred at room temperature for 1 h, and filtered. Cake (2) on the filter paper was washed with 4 L of 70% EtOH in water, and the filtrate and washings were combined to obtain the 70% EtOH extract (a solution containing 233.44 g of solids). This 70% EtOH extract was passed through the resin after washing with 70% EtOH in water. Next, 15 L of 80% EtOH in water (desorption fraction) and 10 L of EtOH (washing fraction) were passed through in that order; the non-adsorbed fraction and the extrusion fraction were combined and dried as the main fraction, yielding RME–CA (rosemary extract–carnosic acid and carnosol).

In addition, cake (2) was collected, 8.5 L of 1.5% sodium hydroxide aqueous solution was added and the mixture was saponified and decomposed in a 70–80 °C water bath while stirring for 2 h and then filtered. Cake (3) on the filter paper was washed with 2.5 L of water and collected. Then, 10 L of 85% EtOH in water was added, stirred for 30 min, and filtered. The obtained filtrate (pH 12) was adjusted to a pH of 4 by adding about 290 mL of 1N hydrochloric acid and the EtOH was distilled off by crude concentration. A total of 1.5 L of water was added to the obtained crude concentrate and the mixture was stirred at room temperature for 1 h in a homomixer and then filtered again. Cake (4) on the filter paper was washed with 1 L of water, collected, and dried to obtain the saponified, decomposed and desalted product as RME–UA (rosemary extract–ursolic acid).

### 2.3. Purification of Carnosic Acid and Carnosol

A total of 19.97 g of RME–CA was added to 100 mL of hexane and stirred at room temperature for 1 h. The mixture was filtered through a Buchner funnel. Residue (1) was washed with 20 mL of hexane. The filtrate was concentrated to obtain filtrate concentrate (1).

A total of 117.86 g of Silica gel 60N was packed in a column with toluene. The filtrate concentrate (1) was dissolved in toluene and charged to the column. Then, 500 mL of toluene, 1000 mL of 3% ethyl acetate/toluene, and 300 mL of 5% ethyl acetate/toluene were passed through in that order. A total of 950 mL of 5% ethyl acetate/toluene was passed through and concentrated to obtain 1.43 g of fraction no. 1. The solution was passed through the column while monitoring with thin-layer chromatography. Fraction no. 1 was dissolved in 20 mL of hexane and cooled at −30 °C for 19 h. After removing as much of the solution as possible, the crystals were concentrated and 1.16 g of carnosic acid crystals were obtained (carnosic acid content: 96.45%).

A total of 16.50 g of the residue (1) was added to 150 mL of toluene and stirred at room temperature for 1 h. The mixture was filtered through a Buchner funnel and washed with 20 mL of hexane. The residue was concentrated to obtain 6.29 g of residue (2). The residue (2) was added to 60 mL of ethyl acetate and stirred at room temperature for 1 h. The mixture was filtered through a Buchner funnel and washed with 20 mL of ethyl acetate. The residue and filtrate were concentrated to obtain 2.95 g of residue (3) and 3.25 g of filtrate concentrate (3).

The residue (3), dissolved in acetone/methanol (1:1), was charged to a column packed with 200.11 g of Silica gel 60N. 500 mL of toluene and 1400 mL of 5% ethyl acetate/toluene were passed through. Then, 1500 mL of 5% ethyl acetate/toluene, and 700 mL of 10% of ethyl acetate/toluene were passed through and concentrated to obtain 0.69 g of fraction no. 2. A total of 800 mL of 10% ethyl acetate/toluene and 900 mL of 20% of ethyl acetate/toluene were passed through and concentrated to obtain 1.12 g of fraction no. 3. A mixture of the filtrate concentrate (3) and the fraction no. 3 dissolved in ethyl acetate/methanol (1:1) was charged to a column packed with 401.40 g of silica gel 60N. A total of 800 mL of toluene, 3200 mL of 5% ethyl acetate/toluene, and 400 mL of 10% of ethyl acetate/toluene were passed through. Then, 1600 mL of 10% ethyl acetate/toluene was passed through and concentrated to obtain 0.79 g of fraction no. 4. The solution was passed through the column while monitoring with thin-layer chromatography. Fraction no. 2 and fraction no. 4 were dissolved in 50 mL of toluene and cooled at 0 °C for 19 h. The mixture was filtered through a Buchner funnel and washed with 20 mL of hexane. The crystals were dissolved and concentrated; 1.23 g of carnosol crystals were obtained (carnosol content: 91.34%).

### 2.4. Expression and Purification of Protein

The full-length SARM1 protein was expressed in ExpiSf9 cells using an ExpiSf Expression System (Thermo Fisher Scientific, Waltham, MA, USA). The DNA region of 1–724 amino acids of human SARM1 with a C-terminal FLAG-tag was inserted into the pFastBac1 vector. The bacmid DNA containing the SARM1 gene was generated by transformation of the pFastBac construct into DH10Bac competent cells. The complex of bacmid DNA and ExpiFectamine Sf Transfection Reagent was added to ExpiSf9 cells. After 5 days of incubation, P0 baculovirus was collected. Next, ExpiSf Enhancer was added to fresh ExpiSf9 cells. 24 h later, the cells were infected with P0 baculovirus and incubated for 3 days. Cells were harvested by centrifugation and stored at −80 °C until use for purification. Harvested cells were lysed by lysis buffer (1% Triton-X100, 20% sucrose, 50 mM Tris–HCl [pH 7.4], 150 mM NaCl, 1% glycerol, and 1 mM EDTA). The resulting lysates were centrifuged to remove insoluble debris; the SARM1-FLAG protein was immunoprecipitated from supernatants by an anti-FLAG M2 affinity gel (Sigma-Aldrich). The immunoprecipitates were washed thoroughly with PBS containing 0.1% Tween-20 (PBS-T) and eluted with 0.5 mg/mL DYKDDDDK peptide (FUJIFILM Wako Pure Chemical). The eluted SARM1 protein was diluted to 1 μM with PBS, aliquoted, and stored at −80 °C until use in the NAD^+^ assay.

A partial sequence of the TIR domain of human SARM1 (594–670 amino acids) fused to glutathione *S*-transferase protein (GST-TIR_594–670_) was purified for the interaction assay with candidate compounds. The recombinant protein was produced in *E. coli* BL21(DE3)pLysS (Promega Bioscience, Madison, WI, USA). Transformed *E. coli* cells were cultured in Luria-Bertani medium containing 100 μg/mL ampicillin at 37 °C. When the optical density of the medium at 600 nm reached ~0.8, 1 mM isopropyl 1-thio-β-D-galactopyranoside was added and the cells were cultured for another 16 h at 25 °C. Harvested *E. coli* cells were lysed by lysis buffer (1% Triton-X100, 20% sucrose, 50 mM Tris–HCl [pH 7.4], 150 mM NaCl, 1% glycerol, 1 mM EDTA) and sonication. The resulting lysates were centrifuged to remove insoluble debris; the GST-TIR_594–670_ protein was purified from the supernatants by a Glutathione Sepharose 4B column (Cytiva, Tokyo, Japan). The supernatants were applied to the column. Non-specifically bound proteins were washed out with PBS-T and the specifically bound GST-TIR_594–670_ protein was eluted with 50 mM Tris–HCl (pH 8.0) containing 10 mM of the reduced form of glutathione. The recovered GST-TIR_594–670_ protein was dialyzed against PBS to remove glutathione, concentrated to 20 μM, aliquoted, and stored at −80 °C until use for the interaction assay.

### 2.5. NAD^+^ Assay

The NAD/NADH-Glo assay (Promega Bioscience) was used to analyze NAD^+^ levels. To analyze SARM1-NADase inhibition by the compounds, 1 μM SARM1 protein in PBS was pre-incubated with the respective compound at a 2% (*v*/*v*) DMSO final assay concentration for 30 min at 37 °C. The reaction was initiated by the addition of 1 μM final concentration of NAD^+^ as a substrate. After 30 min of incubation at 37 °C, the NAD^+^ level was measured by the addition of the NAD/NADH-Glo reagents. Luminescence was observed using a Fluoroskan Ascent FL plate reader (Thermo Fisher Scientific).

To analyze intracellular NAD^+^ levels, a Cell Count Normalization Kit (Dojindo, Kumamoto, Japan) was used to normalize cell numbers before the NAD^+^ assay. The fluorescence intensity of cells stained with Hoechst 33342 was detected by using a FlexStation 3 microplate reader (Molecular Devices, Sunnyvale, CA, USA; Ex: 350 nm; Em: 461 nm). An NAD/NADH-Glo assay was used to analyze NAD^+^ levels. Cells were incubated with NAD/NADH-Glo reagent for 30 min according to the manufacturer’s instructions. The luminescence intensity of cells incubated with NAD/NADH-Glo reagent was detected using a Fluoroskan Ascent FL plate reader (Thermo Fisher Scientific). NAD^+^ levels were normalized by the fluorescence of Hoechst 33342.

### 2.6. Molecular Docking

The chemical structure formula of the compounds was downloaded from the PubChem website. The docking target protein was downloaded from the Protein Data Bank database (7NAK). The MyPresto Portal software (ver. 5) was used to separate the target protein from the original ligand. Then, the compounds and receptors were loaded into docking tools to add polar hydrogens, assign charges, and clean the geometry. A probe point for docking calculation was created from the binding site of the original ligand and molecular docking between the compound and target protein was performed.

### 2.7. Isothermal Titration Calorimeter (ITC)

ITC experiments were performed at 25 °C using a MicroCal iTC_200_ titration calorimeter (Malvern Panalytical, Worcestershire, UK). The buffer for ITC experiments was PBS buffer with 2% DMSO. Titrations were carried out using 2 μL injections applied 4 s apart. The concentration of compounds in the syringe was 10 times the concentration of the protein in the cell: 200 μM of each compound was titrated into 20 μM of GST-TIR_594–670_ protein.

### 2.8. Cell Culture

HEK293T cells and SH-SY5Y cells were cultured in a DMEM/F12 medium (Thermo Fisher Scientific, Waltham, MA, USA) supplemented with 10% fetal bovine serum (FBS). To obtain human-matured neuronal cells in a culture system, 201B7-induced pluripotent stem cells (iPSCs) derived from a healthy 36-year-old female donor (RIKEN BRC, Tsukuba, Japan) were used and were differentiated according to the established protocol, as described previously. Induction of the neural stem cells (NSCs) from iPSCs was conducted using a PSC neural induction medium (Thermo Fisher Scientific) according to the manufacturer’s instructions. After neural induction for 7 days, P0 NSCs were expanded in a neural expansion medium on a dish coated with the Geltrex LDEV-Free hESC-qualified reduced growth factor basement membrane matrix (Thermo Fisher Scientific). To make the Geltrex-coated dish, a dish was incubated with Geltrex matrix solution (diluted 1:100 with Neurobasal medium) for 1 h. For differentiation into neurons, NSCs were cultured at a density of 1 × 10^5^ cells/cm^2^ on a dish coated with 0.002% poly-L-lysine (Sigma-Aldrich) followed by 10 μg/mL laminin (Thermo Fisher Scientific). The culture medium was Neurobasal Plus medium (Thermo Fisher Scientific) supplemented with 2% B-27 plus supplement (Thermo Fisher Scientific), 2 mM GlutaMAX supplement (Thermo Fisher Scientific), CultureOne supplement (Thermo Fisher Scientific), and 200 μM ascorbic acid (Sigma-Aldrich). The spent medium was changed every 3 days and the cells were used for the experiments after 6 days of culture. Cells within passages 5 to 20 were used in the experiments.

### 2.9. Cell Viability Assay

A CellTiter-Glo assay (Promega Biosciences) was used to analyze cell viability. Cells were incubated with CellTiter-Glo detection reagent for 10 min according to the manufacturer’s instructions. Luminescence was observed using a Fluoroskan Ascent FL plate reader (Thermo Fisher Scientific). Cell viability was calculated by setting the control group viability to 100%.

### 2.10. Western Blot Analysis

Western blot analysis was performed under conventional conditions after lysing cells using an SDS sample buffer with PhosphoSTOP (Roche, Indianapolis, IN, USA). The protein concentration of the cell lysates was measured using Pierce 660 nm Protein Assay Reagent (Thermo Fisher Scientific) with Ionic Detergent Compatibility Reagent (Thermo Fisher Scientific). A total of 10 μL of cell lysates was incubated with 150 μL of Pierce 660 nm Protein Assay Reagent for 5 min; the absorbance was measured at 660 nm using an iMark Microplate Reader (Bio-Rad Laboratories, Hercules, CA, USA). A 5 μg aliquot of each protein extract was separated by SDS-polyacrylamide gel electrophoresis and electro-transferred onto an Immobilon membrane (Millipore, Bedford, MA, USA). To detect immunoreactive proteins, HRP-conjugated anti-mouse or anti-rabbit secondary antibodies (CST) and Pierce Western Blotting Substrate Plus (Thermo Fisher Scientific) were used. The blots were placed in contact with FUJI Medical X-ray Film (FUJIFILM, Tokyo, Japan) in a cassette; the light emitted during the chemiluminescent reaction exposed the film. The exposed films were then processed in a darkroom using developing solutions. To quantify the protein levels, the individual band images of proteins were scanned and analyzed using ImageJ software (ver. 1.54g). Then, their intensities were normalized against FLAG or β-Actin as an internal control.

### 2.11. Immunostaining

Cells were incubated with 4% paraformaldehyde for 30 min at room temperature, followed by permeabilization with 1% TritonX-100 in PBS for 20 min, and incubation with a blocking buffer (10% skim milk in PBS-T) for 30 min. Next, the cells were incubated with a mouse NF-L antibody (Santa Cruz Biotechnology; cat#sc-20012, dilution 1:100) and a rabbit β3-tubulin antibody (CST; cat#5568, dilution 1:100) for 18 h at 4 °C. After being washed in PBS-T, the cells were incubated with Alexa Fluor 594 goat anti-mouse IgG antibody (Thermo Fisher Scientific) and Alexa Fluor 488 goat anti-rabbit IgG antibody for 2 h at room temperature. The cells were then washed a second time in PBS-T and mounted using the VECTASHIELD Mounting Medium with DAPI (Vector Laboratories, Burlingame, CA, USA). Finally, the samples were observed under a BZ-X700 fluorescence microscope (Keyence, Osaka, Japan). For the quantification of neurofilament integrity, the fluorescence intensities of NF-L, β3-tubulin, and DAPI were scanned and analyzed using ImageJ software. Then, the fluorescence intensities of NF-L and β3-tubulin were normalized against nuclear numbers.

### 2.12. Animals and Treatments

All experimental procedures were performed in accordance with the Okayama University Advanced Science Research Center’s guidelines for animal experiments and were approved by the Animal Care and Use Committee of Okayama University Advanced Science Research Center. Eight-week-old male C57BL/6J mice were purchased from Charles River Japan (Yokohama, Japan), housed with a 12 h light/dark cycle at a constant temperature (23 °C), and given ad libitum access to food.

After 1 week of habituation, twenty mice were divided into four groups (*n* = 5): the control group, vincristine (VCR) group, VCR + low dose of carnosol group, and VCR + high dose of carnosol group. The sample size was determined based on previous studies to obtain a significance level. VCR was dosed intraperitoneally at 1 mg/kg in saline daily for 5 days. Carnosol was dosed orally by gavage at 10 and 100 mg/kg in vehicle (5% DMSO, 10% methyl-β-cyclodextrin) daily starting 2 h before VCR. The control group received only the solvent. On Day 8 after the start of administration, a von Frey mechanical threshold test, plasma collection, and tissue collection were performed. In another study, RME–CA was used instead of carnosol, with all other conditions remaining the same.

### 2.13. Von Frey Mechanical Threshold Test

Mice were placed in a transparent plexiglass box on an elevated wire mesh grid and allowed to habituate to the testing environment. A series of standardized von Frey fibers was used to stimulate the mouse hind paw with the up–down paradigm in sequential order (Ugo Basile, Gemonio, Italy). A higher-force filament was tested if the mouse had no withdrawal response, while a lower-force filament was used if there was a withdrawal response. After the first change in direction, four more tests were performed.

### 2.14. Plasma Phosphorylated Neurofilament H (pNF-H) Measurement

To measure the amount of pNF-H released into plasma, plasma was diluted 1:5 in Elisa Assay Diluent (Merck Millipore, Burlington, MA, USA) and pNF-H was measured using the pNF-H sandwich ELISA kit assay (Merck Millipore) according to the manufacturer’s instructions.

### 2.15. Quantification of Intraepidermal Nerve Fibers

Plantar skin specimens were excised from the hind paw under isoflurane anesthesia. Specimens were immediately fixed for 24 h in Zamboni solution (FUJIFILM Wako Pure Chemical) and cryopreserved for 24 h in PBS with 30% sucrose with gentle shaking. Specimens were then embedded in Tissue-Tek OCT compound (Sakura Finetek, Tokyo, Japan), frozen, and sliced into 20 μm sections on a cryostat. The sections were washed with PBS, permeabilized with PBS-T, and incubated with a blocking solution consisting of 5% normal goat serum in PBS. The sections were stained with antibodies to PGP9.5 (1:1000) followed by fluorescently labeled secondary antibodies. After being washed with PBS-T, the sections were mounted using VECTASHIELD Mounting Medium with DAPI (Vector Laboratories). The sections were observed under an LSM780 confocal laser-scanning microscope (Carl Zeiss, Jena, Germany). The density of the intraepidermal nerve fibers (IENF) was determined by counting the number of IENFs crossing the dermal–epidermal junction in three randomly selected slices per footpad. The length of the epidermis was measured using ImageJ software and the numbers of the IENFs (fiber numbers/mm) were obtained.

### 2.16. Statistical Analysis

Prior to statistical analysis, each experiment was repeated three times. The results are expressed as means ± S.D. All statistical analyses were performed with EZR (Saitama Medical Center, Jichi Medical University, Saitama, Japan), which is a graphical user interface for R (The R Foundation for Statistical Computing, Vienna, Austria). One-way and two-way ANOVA were used for comparison. If the ANOVA showed a significant difference, Tukey’s test was used as a post hoc test. *p* values of less than 0.05 were considered statistically significant.

## 3. Results

### 3.1. Carnosol, an Antioxidant Ingredient of Rosemary, Inhibits the NAD^+^-Cleavage Activity of SARM1

SARM1 responds to stress, induces axonal degeneration by hydrolyzing NAD^+^, and is involved in the pathological progression of various neurological diseases. To search for compounds that inhibit the NAD^+^-cleavage activity of SARM1, full-length SARM1 tagged with FLAG was overexpressed in ExpiSf9 insect cells and was purified using an anti-FLAG-affinity gel (Figure 1A). We previously reported that c-Jun N-terminal kinase (JNK)-mediated phosphorylation of SARM1-Ser548 increases NAD^+^-cleavage activity [18]. It was confirmed that SARM1 expressed in ExpiSf9 cells had a significantly higher phosphorylation level at Ser548 compared to SARM1 expressed in HEK293T animal cells (Figure 1B, left). Because the two cell lines are different species (human and insect origins), for which the general antibodies for detecting housekeeping proteins as measuring loading control levels will not react equally, CBB-stained bands in the gel were used as a loading control (Figure 1B, right). The purified full-length SARM1 had NAD^+^-cleavage activity (Figure 1C). As a result of a preliminary search for compounds that inhibit the enzymatic activity of SARM1 from food ingredients, it was found that rosemary extract has SARM1 inhibitory activity. Therefore, the rosemary components were extracted into three fractions, which were labeled RME–RA, RME–CA, and RME–UA; it was found that RME–CA was responsible for the SARM1 inhibitory activity (Figure 1C). The RME–RA fraction of rosemary mainly contained rosmarinic acid; the RME–CA fraction mainly contained carnosic acid and carnosol; and the RME–UA fraction mainly contained ursolic acid. Carnosic acid and carnosol were purified from the RME–CA fraction (Figure 1D) and further analyzed. When their inhibitory effects on the NAD^+^-cleavage activity of SARM1 were compared, carnosol inhibited SARM1 at lower concentrations than carnosic acid (Figure 1E). Purified rosmarinic acid and ursolic acid did not have SARM1 inhibitory activity (Figure 1F). These results showed that the carnosol in rosemary extract plays a central role in SARM1 inhibitory activity, suggesting that it may have potential for the treatment of neurological diseases.

### 3.2. Carnosol Interacts with the TIR Domain of SARM1

To determine how carnosol inhibits the NAD^+^-cleavage activity of SARM1, we analyzed the interaction between the compound and SARM1 using a molecular docking tool and an isothermal titration calorimeter. The SARM1-TIR domain data (7NAK) registered in the Protein Data Bank were used as the target molecule for compound docking calculations. The molecular-docking analysis indicated that carnosol has a stronger interaction with the NAD^+^-binding pocket of the TIR domain than carnosic acid (Figure 2A–C). Next, an isothermal titration calorimeter was used to analyze the direct interaction between the compound and a partial sequence of the TIR domain (594–670 amino acids) fused to the GST protein (GST-TIR_594–670_). Although we attempted to purify the entire TIR domain, it was difficult to purify it at high concentrations; therefore, a partial sequence of the TIR domain was used. While carnosic acid showed little interaction with GST-TIR_594–670_, carnosol showed a strong interaction with GST-TIR_594–670_ (Figure 2D–F). These results suggest that carnosol binds to the NAD^+^ binding pocket of the SARM1-TIR domain and acts competitively with NAD^+^, inhibiting the cleavage of NAD^+^ by SARM1.

### 3.3. Carnosol Suppresses SARM1-Dependent Cytotoxic Activity

Carnosic acid and carnosol have been shown to exert neuroprotective effects through antioxidant phase 2 enzyme induction initiated by activation of the KEAP1/NRF2 transcription pathway [42,43]. Therefore, we next examined whether carnosol suppresses cell death by directly inhibiting SARM1. The N-terminal ARM domain of SARM1 functions as an autoinhibitory domain; overexpression of SARM1_409–724_ without this domain reduces cell viability. SARM1_409–724_ was overexpressed in HEK293T cells and cell viability was measured by adding carnosic acid and carnosol. Carnosic acid could hardly suppress SARM1-induced cell death, but carnosol suppressed it in a concentration-dependent manner (Figure 3A). SARM1 inhibits mitochondrial respiration by consuming NAD^+^, decreasing ATP levels. Therefore, overexpression of SARM1 enhances AMP-activated protein kinase (AMPK) phosphorylation, a process that regulates cellular energy balance, by increasing the ADP/ATP ratio. In parallel with the improvement in cell viability, carnosol suppressed AMPK phosphorylation, whereas carnosic acid did not (Figure 3B). Recently, it has been reported that the rodenticide Vacor precisely activates SARM1 in neurons, inducing axonal degeneration and cell death [47,48]. When Vacor, carnosic acid, and carnosol were added to human neuroblastoma SH-SY5Y or human iPSC-derived neurons, carnosol suppressed Vacor-induced cell death (Figure 3C,D). These results indicate that carnosol inhibits SARM1 directly and suppresses SARM1-dependent cytotoxic activity.

### 3.4. Carnosol Suppresses Vincristine-Induced Neurite Degradation and Cell Death

Next, the neuroprotective effect of carnosol using neurons differentiated from human iPSCs was examined. Anticancer drugs, such as vincristine (VCR), exert antitumor effects by inhibiting cell division but they are also known to act on microtubules in neurons, inducing axon degeneration and cell death [49]. VCR induced neurite degradation and reduced cell viability in iPSC-derived neurons but carnosol significantly inhibited these phenomena (Figure 4A,B). VCR activates SARM1 in neurons, reduces NAD^+^ levels, and induces axon degeneration [49]. VCR treatment reduced the NAD^+^ levels in neurons; however, carnosol was able to inhibit this effect (Figure 4C). At the molecular level, VCR induced the degradation of the axonal components neurofilament-M (NF-M) and neurofilament-L (NF-L) and increased the amount of cleaved caspase 3 (Cl. Caspase 3), an apoptosis marker (Figure 4D). Carnosol inhibited these changes. The protein level of SARM1 was not affected by the addition of VCR or carnosol. Morphological observation showed that VCR induced axonal degradation; however, carnosol again blocked this change (Figure 4E–G). A similar improvement was observed when fraction RME–CA of the rosemary extract was used (Appendix A). Although rosmarinic acid and ursolic acid are known to have neuroprotective effects, these compounds did not inhibit VCR-induced cell death (Figure 4H). Collectively, these results indicate that carnosol inhibits VCR-induced neurite degradation and cell death through the inhibitory effect of SARM1.

### 3.5. Carnosol Ameliorates Symptoms of Vincristine-Induced Peripheral Neuropathy

To confirm the neuroprotective effect of carnosol via SARM1 in vivo, we performed an analysis using a VCR-induced peripheral neuropathy model. It has been reported that the symptoms of VCR-induced peripheral neuropathy are significantly reduced in SARM1 knockout mice [24]. Carnosol was orally administered at 0–100 mg/kg and VCR was intraperitoneally administrated at 1 mg/kg 2 h later. This administration protocol was continued for 5 days. Three days after completing the carnosol administration (i.e., on day 8 of the experiment), various analyses were performed (Figure 5A). Von Frey’s study showed that the administration of VCR caused hyperalgesia in mice. Carnosol reduced this symptom in a concentration-dependent manner (Figure 5B). VCR acts on microtubules to destroy axonal components and release them into the blood. VCR administration released phosphorylated neurofilament-H (pNF-H), an axonal component, into the blood; this release was inhibited by carnosol (Figure 5C). To analyze histological changes, the state of plantar intraepidermal nerve fibers (IENFs) was analyzed using a PGP9.5 antibody. VCR administration reduced the density of IENFs; however, carnosol inhibited this reduction (Figure 5D,E). A similar improvement was observed when fraction RME–CA of the rosemary extract was used (Appendix A). These results indicate that carnosol effectively alleviates the symptoms of VCR-induced peripheral neuropathy.

## 4. Discussion

This is the first report to show that carnosol, an antioxidant component in rosemary, is a potent inhibitor of SARM1, and, therefore, a powerful suppressor of SARM1-induced axon degeneration and cell death. At the organism level, we found that carnosol potently counteracted VCR-mediated hyperalgesia symptoms in a mouse model.

We have studied SARM1 in neurodegeneration because of its unusual role as a neuronal cell executioner in response to several neuro-stresses where SARM1 is phosphorylated at Ser548, which is important for SARM1 activation [18,50,51]. It is conceivable that the phosphorylation or enzymatic activity of SARM1 could be specifically targeted in order to regulate various types of neurodegenerative diseases caused by a wide range of stressors. Although the inhibition of JNK is associated with the Ser548 phosphorylation inhibition of SARM1, the inhibitory strategy of JNK never specifies the SARM1 protein. Therefore, we propose a catalytic suppression tactic, which involves directly targeting the enzymatic activity of SARM1. The point is that SARM1 expression is relatively limited to neuronal cells; therefore, the candidate medicine will not be prevalent in other cells, which should help to prevent unexpected side effects.

Based on this concept, we discovered that rosemary, a food ingredient, inhibits SARM1. In this study, we screened 22 food ingredients for SARM1 inhibitory activity; however, rosemary was the only food ingredient that clearly showed SARM1 inhibitory activity. Further fractionation of rosemary extract revealed that carnosol has SARM1 inhibitory activity. Carnosic acid, which has a structure very similar to carnosol, also showed SARM1 inhibitory activity when used at high concentrations. Since other rosemary ingredients did not show SARM1 inhibitory activity, even at high concentrations, the diterpene structure is likely essential for interaction with the TIR domain of SARM1. When we investigated the SARM1 inhibitory activity of several other compounds with diterpene structures, dihydroabietylamine showed SARM1 inhibitory activity. Based on these findings, it may be possible to synthesize compounds that more strongly inhibit SARM1 by synthesizing various derivatives based on the diterpene skeleton.

The purpose of this study was to find a beneficial compound that inhibits axonal degeneration through SARM1 inhibition and alleviates the symptoms of various neurological diseases from food ingredients. Rosemary and carnosol have been known to have neuroprotective effects through their antioxidant and anti-inflammatory properties; however, there have been no clear reports that they directly inhibit axonal degeneration. Therefore, this study is significant in that it has discovered a new function of carnosol that alleviates neurological symptoms by inhibiting axonal degeneration. The standard treatments for neurological diseases are relatively expensive and may cause side effects. However, the potential cost-effectiveness of rosemary, which contains carnosol and other neuroprotective ingredients that can be ingested on a daily basis, shows new possibilities for the prevention and treatment of neurological diseases. Carnosol is found not only in rosemary but also in plants of the Lamiaceae family, such as sage; thus, the use of these plants is also thought to be effective in alleviating neurological symptoms.

Despite the many advances in cancer treatment, the pain and peripheral neuropathy caused by anti-cancer drugs remains a major problem. Unfortunately, there is still no clear choice for a safe, potent, and effective pain reliever for use during chemotherapy. In our animal model in this study, carnosol usage did lighten the pain caused by VCR treatment by protecting peripheral neuronal cells without appreciable body weight changes. These results suggest that SARM1 inhibition by carnosol is a promising avenue for suppressing CIPN. Meanwhile, symptom management is also essential in peripheral neuropathy. In addition to pharmacological approaches, such as carnosol, the integration of therapeutic exercises and comprehensive rehabilitation programs holds great potential. These exercises may help to prevent disability and support the maintenance of an adequate quality of life, as both composite approaches have the potential to accelerate the regeneration of complex neural networks under the fixed protection of neuronal axons [52].

Symptoms of pain and peripheral neuropathy tend to increase with age; the decrease in NAD^+^ levels due to activation of SARM1 may be involved in these symptoms. In the context of anti-aging medicines, much attention has been paid to the NAD^+^ levels in neuronal cells, the dysregulation of which has been associated with senescence-associated diseases such as dementia. Some supplements, such as nicotinamide mononucleotide (NMN), are often used to supply the metabolic product NAD^+^ to cells in the body [53,54]. However, in the aged brain and peripheral nervous system, SARM1 is activated, which may cause the supplied NAD^+^ to be rapidly decreased. In addition, it has also been found that NMN itself activates SARM1 [17]. The potential of NMN to ameliorate the symptoms of aging is an intriguing area for future research. Indeed, one day, a cocktail of carnosol and NMN may prevent or even reverse some of the unwanted effects of age.

A limitation of this study is that it is unclear to what extent the in vivo improvement effect of carnosol on peripheral neuropathy depends on SARM1 inhibition. Although it was confirmed in vitro that carnosol suppresses SARM1-dependent cell death, it is thought that in vivo, the symptoms were alleviated by a combination of factors. This suggests that further research is needed to fully understand the complex mechanisms. Indeed, carnosol alleviates the symptoms of peripheral neuropathy through not only SARM1 inhibition, but also through an antioxidant effect via NRF2 activation. By comparing with carnosic acid, which has an NRF2 activation effect but low SARM1 inhibitory activity, it may be possible to more clearly demonstrate the improvement effect of carnosol through its SARM1 inhibitory activity.

In conclusion, our original screening of food ingredients has led to a significant discovery. Carnosol, a component of rosemary, enables potent inhibition of the NAD^+^-cleavage activity of SARM1, a novel finding in the field. This inhibition, in turn, suppresses axon degeneration and cell death induced by the anti-cancer medicine VCR and also alleviates VCR-induced hyperalgesia symptoms. The fact that rosemary contains a variety of neuroprotective compounds in addition to carnosol proposes it as a superior food herb. This study not only demonstrates the potential of rosemary and carnosol as therapeutic agents for CIPN but also makes a substantial contribution to the treatment of various neurological diseases. It underscores the promising future of SARM1 as a target for future therapeutic interventions.

## Figures and Tables

**Figure 1 antioxidants-14-00808-f001:**
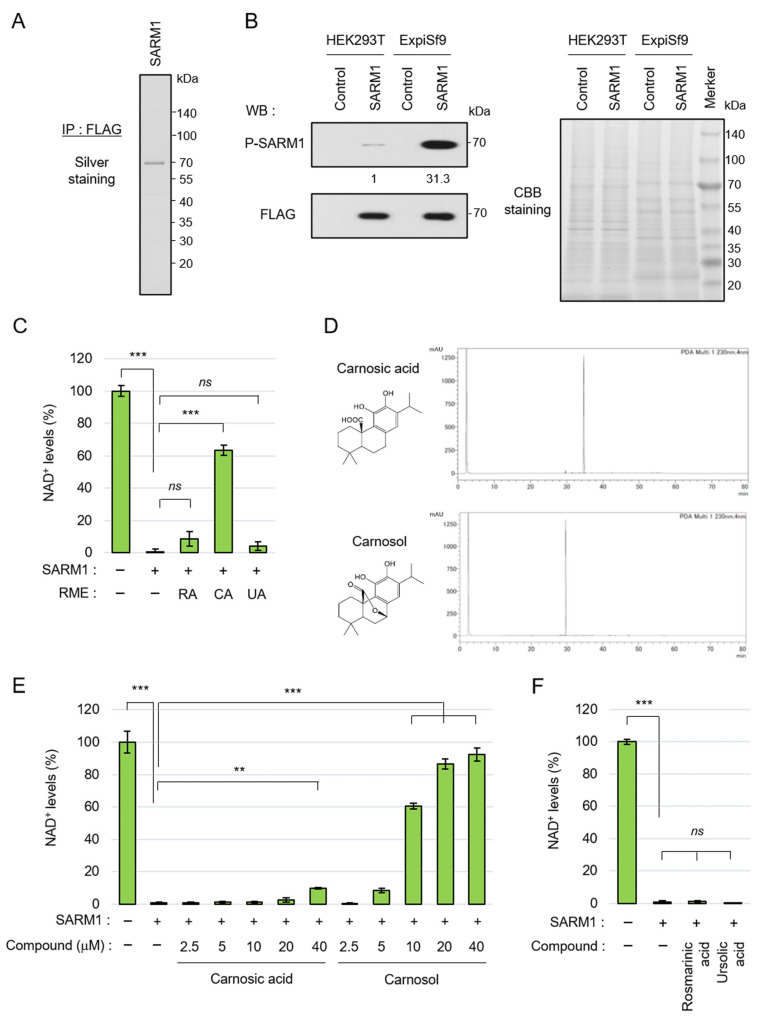
Carnosol inhibited SARM1-NADase activity: (**A**) silver staining gel of purified SARM1 protein. A full-length SARM1-FLAG protein was purified from lysates of ExpiSf9; (**B**) comparison of phosphorylation levels of SARM1 expressed in HEK293T and ExpiSf9. The band intensities of P-SARM1 were normalized to that of FLAG and the values are indicated below the panel. A CBB-stained gel of cell lysates was used as a loading control; (**C**) inhibition analysis of SARM1-NADase using rosemary extracts. A total of 1 μM SARM1 protein was incubated with 10 μg/mL RME–RA, RME–CA, or RME–UA; (**D**) HPLC data of purified carnosic acid and carnosol from rosemary extracts; (**E**) inhibition analysis of SARM1-NADase using carnosic acid and carnosol. A total of 1 μM SARM1 protein was incubated with 0~40 μM of the compounds; and (**F**) rosmarinic acid and ursolic acid did not have SARM1 inhibitory activity. A total of 1 μM SARM1 protein was incubated with 10 μM compounds. *ns*: not significant; ** *p* < 0.01; *** *p* < 0.001.

**Figure 2 antioxidants-14-00808-f002:**
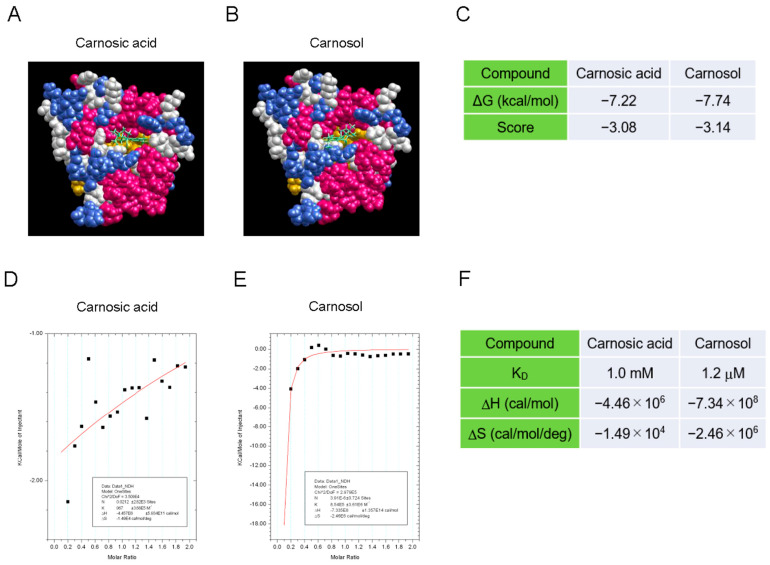
Interaction between the SARM1-TIR domain and carnosol (**A**–**C**). An in silico assay of the interaction between the SARM1-TIR domain and compounds by myPresto Portal. The cryo-EM structure of activated human SARM1 in a complex with NMN and 1AD (7NAK; Protein Data Bank) was used as the receptor molecule: (**A**) carnosic acid (PubChem CID: 65126); (**B**) carnosol (PubChem CID: 442009) were used as docking compounds of the SARM1-TIR domain; and (**C**) the docking score of compounds to the SARM1-TIR domain. (**D**,**F**) Analysis of the direct interaction between SARM1_594–670_ and compounds using isothermal titration calorimetry. 20 μM GST-fused SARM1_594–670_ was used as the receptor molecule: (**D**) 200 μM carnosic acid; (**E**) 200 μM carnosol were used as ligands of SARM1_594–670_; and (**F**) the ability of the tested compounds to interact with GST-SARM1_594–670_.

**Figure 3 antioxidants-14-00808-f003:**
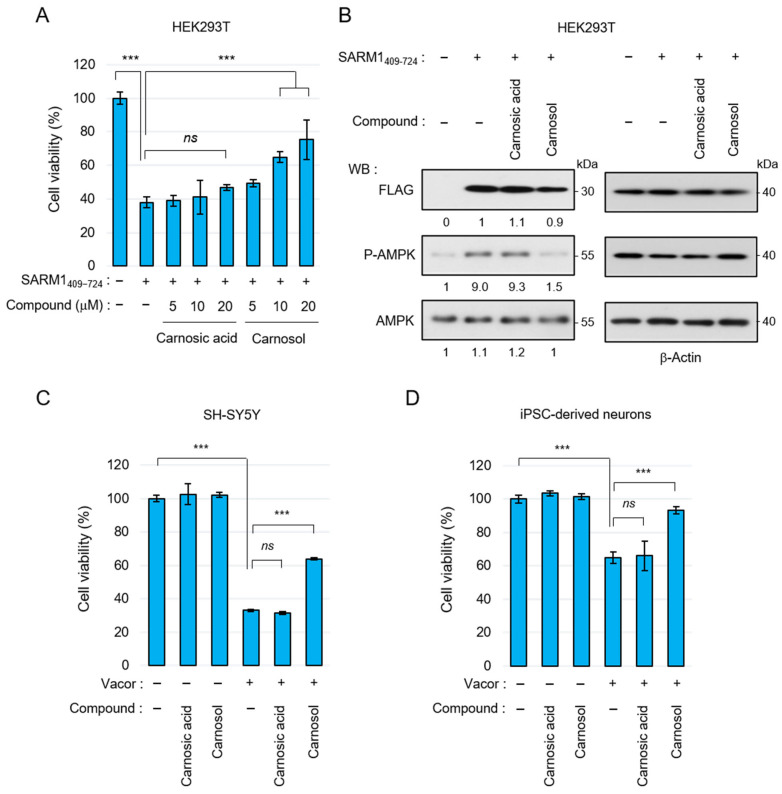
Suppression of SARM1-dependent cytotoxic activity by carnosol: (**A**) relative cell viability of HEK293T cells expressing SARM1_409–724_ treated with 0~20 μM carnosic acid or 0~20 μM carnosol for 16 h; (**B**) carnosol suppressed the AMPK phosphorylation induced by SARM1_409–724_. HEK293T cells expressing SARM1_409–724_ were treated with DMSO (control), 10 μM carnosic acid, or 10 μM carnosol for 16 h. The band intensities of the indicated proteins were normalized to their corresponding bands of β-Actin in the right panel and the values are indicated below each panel; (**C**) suppression of vacor-induced cell death by carnosol in SH-SY5Y cells. SH-SY5Y cells were treated for 8 h with 50 μM vacor, 10 μM carnosic acid, or 10 μM carnosol; and (**D**) suppression of vacor-induced cell death by carnosol in iPSC-derived neurons. Human iPSC-derived neurons were treated for 8 h with 10 μM vacor, 10 μM carnosic acid, or 10 μM carnosol. *ns*: not significant; *** *p* < 0.001.

**Figure 4 antioxidants-14-00808-f004:**
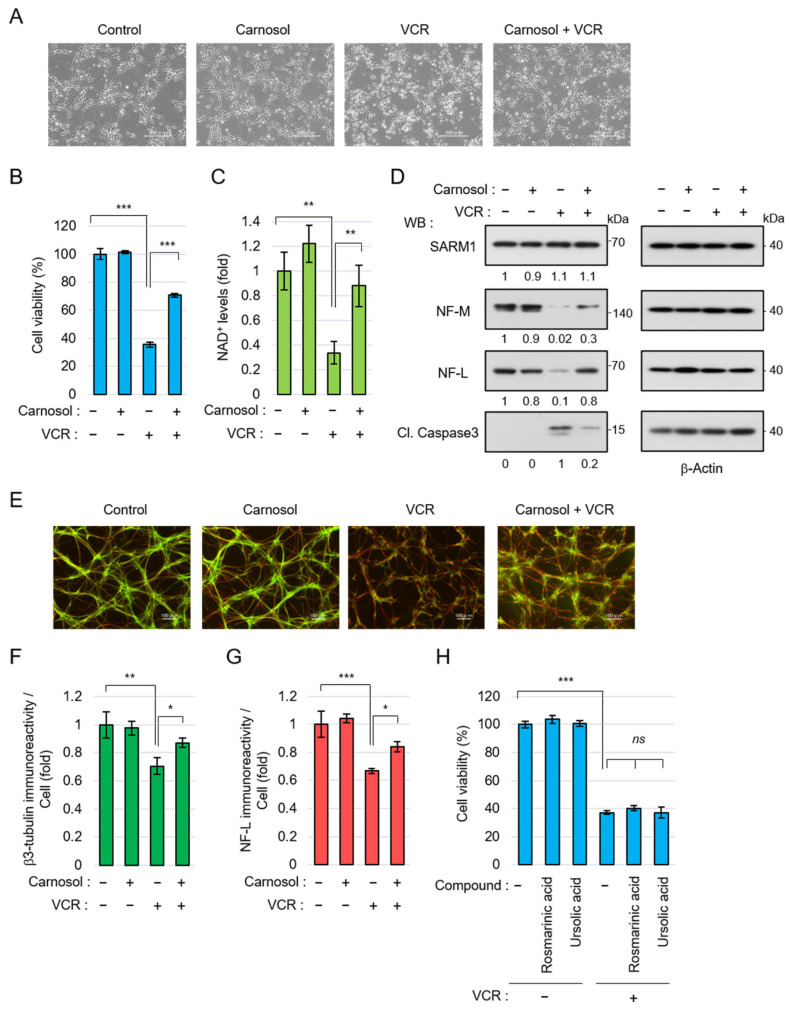
Carnosol suppressed vincristine (VCR)-induced neurite degeneration and cell death: (**A**–**G**) human iPSC-derived neurons were treated with 50 nM VCR and 10 μM carnosol for 24 h (**A**–**D**) or 8 h (**E**–**G**). Panels (**A**–**D**) show, as follows: (**A**) cell morphologies, scale bar: 200 μm; (**B**) cell viability; (**C**) NAD^+^ levels; and (**D**) Western blotting results for cell lysates at 24 h after the treatment. The band intensities of the indicated proteins were normalized to their corresponding bands of β-Actin in the right panel and the values are indicated below each panel. (**E**) Representative images of neuronal axons after 8 h of treatment with the indicated compounds. Stain: anti-β3 tubulin and NF-L. Scale bar: 100 μm; (**F**,**G**) immunoreactivity of β3-tubulin and NF-L. The fluorescence intensities of β3-tubulin and NF-L were normalized against nuclear numbers; and (**H**) treatment with rosmarinic acid or ursolic acid did not suppress VCR-induced cell death. The cell viabilities of iPSC-derived neurons that were treated for 24 h with 50 nM VCR, 10 μM rosmarinic acid, or 10 μM ursolic acid. *ns*: not significant, * *p* < 0.05, ** *p* < 0.01, *** *p* < 0.001.

**Figure 5 antioxidants-14-00808-f005:**
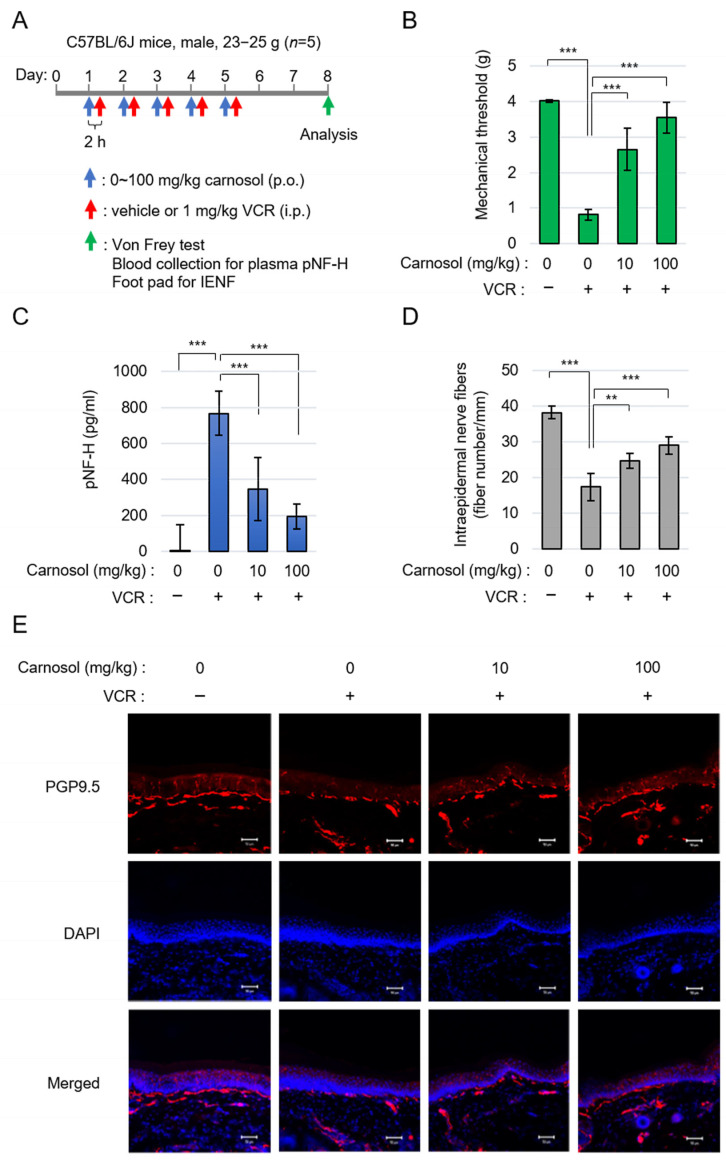
Carnosol treatment attenuated VCR-induced peripheral neuropathy: (**A**) schematic diagram of the treatment. (**B**) quantification of mechanical sensitivity; (**C**) quantification of plasma pNF-H levels; and (**D**,**E**) the changes in INEF density in mice given the treatment. Quantification data (**D**) and the representative figures (**E**). Stain: anti-PGP9.5 and DAPI. Scale bar: 50 μm. ** *p* < 0.01, *** *p* < 0.001.

## Data Availability

Data will be made available on request.

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
