# Peer review of "Carnosol, a Rosemary Ingredient Discovered in a Screen for Inhibitors of SARM1-NAD+ Cleavage Activity, Ameliorates Symptoms of Peripheral Neuropathy"

_antioxidants, 2025, doi:10.3390/antiox14070808_

Round 1
Reviewer 1 Report
Introduction should be enriched with recent works supporting the action of Rosemary and Carnosol in neurons. Discussion is poorly presented, authors should consider the reported advances in the field. The effects of structurally related compounds and the clear support of the selection of carnosol to be tested as neuroprotective agent. Hence, the results should be discussed as other compounds have been tested as active in this evaluations, moreover, the effects of other compounds in Rosemary and the addition of positive control acting in the evaluated systems is desirable. Not just comparison with the absence of treatment in some evaluations. Limitations should be declared.
Conclusions should be clear in a final section or in the end of the discussion clearly supported considering all results and discussion.
The sentence in Page 2 :
The medicinal potential of rosemary in preventing devastating neurodegenerative diseases such as Alzheimer's disease and Parkinson's disease (32) was a source of inspiration and motivation for our present research.
should be as regular letters (is in bold).
Author Response
Comment 1: Introduction should be enriched with recent works supporting the action of Rosemary and Carnosol in neurons.
Response: We thank the reviewer for the comment. We have mentioned the involvement of SARM1 in neurodegeneration with recent works in introduction: “The deletion of SARM1 has also been reported to alleviate symptoms of Alzheimer's disease, Parkinson's disease, and ALS, suggesting that it may be effective in treating neurodegenerative diseases (27-29).”, and added the action of Rosemary and Carnosol in neurons with previous works in introduction: “Rosemary has been reported for its unusual druggable role in neuronal protection via multiple mechanisms. For instance, it suppresses the amyloid beta-induced decline in mitochondrial membrane potential and reduces rotenone-induced lipid peroxidation and oxidative stress (35,36). Attributing to the medicinal-gifts trait of the herb as mentioned above, it has been studied for its potential in treating some neurodegenerative diseases, culminating in showing beneficial effects on Alzheimer’s disease and Parkinson’s disease (37). What is the medicinal nature of rosemary in these intractable diseases? The neuroprotective components of rosemary that have been reported are as follows: rosmarinic acid, ursolic acid, carnosic acid, and carnosol.”, “In addition, rosmarinic acid has been shown to exhibit neuroprotective effects on glutamatergic synapses by regulating oxidative phosphorylation in MPTP-induced Parkinson's disease model mice, as demonstrated by a proteomic approach in the substantia nigra of the disease model (40).”, “NRF2 normally binds to Kelch-like ECH-associated protein 1 (KEAP1) and its expression level is kept low, but carnosic acid and carnosol inhibit KEAP1 and activate NRF2 (44,45). Activation of NRF2 up-regulates gene expression of antioxidant and detoxification enzymes, protecting neurons from oxidative stress and inflammation. Carnosol has also been shown to improve protein homeostasis and mitochondrial homeostasis, and reduce pathogenic protein aggregation and cognitive impairment (46).”
Comment 2: Discussion is poorly presented, authors should consider the reported advances in the field. The effects of structurally related compounds and the clear support of the selection of carnosol to be tested as neuroprotective agent. Hence, the results should be discussed as other compounds have been tested as active in this evaluations, moreover, the effects of other compounds in Rosemary and the addition of positive control acting in the evaluated systems is desirable. Not just comparison with the absence of treatment in some evaluations.
Response: Thank you for your thoughtful comment. We agree that a more in-depth discussion of carnosol and the structurally related compounds would enhance the significance and clarity of our findings. We have added in discussion: “Based on this concept, we discovered rosemary as a food ingredient that inhibits SARM1. In this study, we screened 22 food ingredients for SARM1 inhibitory activity, but rosemary was the only food ingredient that clearly showed SARM1 inhibitory activity. Further fractionation of rosemary extract revealed that carnosol has SARM1 inhibitory activity. Carnosic acid, which has a structure very similar to carnosol, also showed SARM1 inhibitory activity when used at high concentrations. Since other rosemary ingredients did not show SARM1 inhibitory activity even at high concentrations, the diterpene structure is likely essential for interaction with the TIR domain of SARM1. When we investigated the SARM1 inhibitory activity of several other compounds with diterpene structures, dihydroabietylamine showed SARM1 inhibitory activity (data not shown). Based on these findings, it may be possible to synthesize compounds that more strongly inhibit SARM1 by synthesizing various derivatives based on the diterpene skeleton.
The purpose of this study was to find a beneficial compound that inhibits axonal degeneration through SARM1 inhibition and alleviates the symptoms of various neurological diseases from food ingredients. Rosemary and carnosol have been known to have neuroprotective effects through their antioxidant and anti-inflammatory properties, but there have been no clear reports that they directly inhibit axonal degeneration. Therefore, this study is significant in that it has discovered a new function of carnosol that alleviates neurological symptoms by inhibiting axonal degeneration. The standard treatments for neurological diseases are relatively expensive and may cause side effects. However, the potential cost-effectiveness of rosemary, which contains carnosol and other neuroprotective ingredients that can be ingested on a daily basis, shows new possibilities for the prevention and treatment of neurological diseases. Carnosol is found not only in rosemary but also in plants of the Lamiaceae family, such as sage, so the use of these plants is also thought to be effective in alleviating neurological symptoms”.
Comment 3: Limitations should be declared.
Response: We thank the reviewer for the comment. We have added a limitation of this study in discussion: “A limitation of this study is that it is unclear to what extent the in vivo improvement effect of carnosol on peripheral neuropathy depends on SARM1 inhibition. Although it was confirmed in vitro that carnosol suppresses SARM1-dependent cell death, it is thought that in vivo the symptoms were alleviated by a combination of factors. This suggests that further research is needed to fully understand the complex mechanisms. Indeed, carnosol alleviates the symptoms of peripheral neuropathy through not only SARM1 inhibition, but also antioxidant effect via NRF2 activation. By comparing with carnosic acid, which has NRF2 activation effect but low SARM1 inhibitory activity, it may be possible to more clearly demonstrate the improvement effect of carnosol through its SARM1 inhibitory activity.”
Comment 4: Conclusions should be clear in a final section or in the end of the discussion clearly supported considering all results and discussion.
Response: We appreciate your feedback. We have added conclusions of this study in the end of the discussion: “In conclusion, our original screening of food ingredients has led to a significant discovery. Carnosol, a component of rosemary, enables potent inhibition of the NAD+-cleavage activity of SARM1, a novel finding in the field. This inhibition, in turn, suppresses axon degeneration and cell death induced by the anti-cancer medicine VCR and also alleviates VCR-induced hyperalgesia symptoms. The fact that rosemary contains a variety of neuroprotective compounds in addition to carnosol proposes it as a superior food herb. This study not only demonstrates the potential of rosemary and carnosol as therapeutic agents for CIPN but also makes a substantial contribution to the treatment of various neurological diseases. It underscores the promising future of SARM1 as a target for future therapeutic interventions.”
Comment 4: The sentence in Page 2 should be as regular letters (is in bold).
Response: We have deleted this sentence.
Reviewer 2 Report
The study aims to demonstrate that carnosol, a compound found in rosemary, ameliorates symptoms of peripheral neuropathy through SARM1 inhibition. Carnosol, known for its antioxidant properties, inhibits the NAD⁺-cleaving activity of SARM1.
TITLE: Please include the study design in the title.
MAIN TEXT: Use a more formal academic tone by avoiding first-person language such as “we” and “our.”
LIMITATIONS: Please include a section outlining the study’s limitations.
DISCUSSION: Consider emphasizing the following point:
Symptom management is essential in peripheral neuropathy. The integration of therapeutic exercises and rehabilitation programs may help prevent disability and support the maintenance of an adequate quality of life (DOI: 10.3390/diagnostics10121022).
none
Author Response
Comment 1: TITLE: Please include the study design in the title.
Response: We thank the reviewer for the comment. The title has changed including the study design.
Comment 2: MAIN TEXT: Use a more formal academic tone by avoiding first-person language such as “we” and “our”.
Response: We appreciate the reviewer’s advice, and the main text has revised by avoiding first-person language as much as possible.
Comment 3: LIMITATIONS: Please include a section outlining the study’s limitations.
Response: We thank the reviewer for the comment. We have added a limitation of this study in discussion: “A limitation of this study is that it is unclear to what extent the in vivo improvement effect of carnosol on peripheral neuropathy depends on SARM1 inhibition. Although it was confirmed in vitro that carnosol suppresses SARM1-dependent cell death, it is thought that in vivo the symptoms were alleviated by a combination of factors. This suggests that further research is needed to fully understand the complex mechanisms. Indeed, carnosol alleviates the symptoms of peripheral neuropathy through not only SARM1 inhibition, but also antioxidant effect via NRF2 activation. By comparing with carnosic acid, which has NRF2 activation effect but low SARM1 inhibitory activity, it may be possible to more clearly demonstrate the improvement effect of carnosol through its SARM1 inhibitory activity.”
Comment 4: DISCUSSION: Consider emphasizing the following point.
Response: Thank you for your thoughtful comment. We agree that symptom management is essential in peripheral neuropathy, and have added in discussion: “Meanwhile, symptom management is also essential in peripheral neuropathy. In addition to pharmacological approaches such as carnosol, the integration of therapeutic exercises and comprehensive rehabilitation programs holds great potential. These exercises may help prevent disability and support the maintenance of an adequate quality of life, as both composite approaches have the potential to accelerate the regeneration of complex neural networks under the fixed protection of neuronal axons (52).”
Reviewer 3 Report
The manuscript by Hitoshi Murata et al., entitled “Carnosol, a rosemary ingredient, ameliorates symptoms of peripheral neuropathy through SARM1 inhibition” reported that the treatment of carnosol inhibited the enzymatic activity of SARM1 and suppressed neurite degeneration and cell death induced by the anti-cancer medicine vincristine (VCR). Carnosol inhibits VCR-induced hyperalgesia symptoms, suppressed the loss of intraepidermal nerve fibers in vivo, and reduces the blood fluid level of phosphorylated neurofilament-H caused by an axonal degeneration event. The authors conclude carnosol has a neuroprotective effect via SARM1 inhibition in addition to its antioxidant effect via NF-E2 related factor 2 and suppresses neurotoxin-induced peripheral neuropathy. The authors confirmed the results by various techniques. The manuscript is well-written. However, some remarks should be taken by the authors into consideration.
Comments:
- For the western blot analysis, the protein concentration was measured? Please include the method details.
- The authors should include the detailed procedure for WB imaging and image quantification methods.
- Please provide more details regarding the animals and animal number used in the study. There is no description of the total number of mice used for the study, N per experimental group, or N per analyzed endpoint in the Methods.
- How was the sample size determined?
- In immunostaining methods, the authors should include the details about image quantification,
- In statistical analysis section, the authors replace the +/- with “±” symbol.
- In Figure 1B, the authors should represent the loading control protein expression.
- In Figure 3B and 4D, the WB images were from the same membrane? If not please represent individual loading control for each WB images.
- In Figure 5E, the authors should include the merged images of PGP9.5 and DAPi.
- How the intraepidermal nerve fiber numbers was counted? Please include the details in materials and method section.
Author Response
Comment 1: For the western blot analysis, the protein concentration was measured? Please include the method details.
Response: We thank the reviewer for the comment. Yes, the protein concentration was measured. We have added in materials and methods: “The protein concentration of the cell lysates was measured using Pierce 660 nm Protein Assay Reagent (Thermo Fisher Scientific) with Ionic Detergent Compatibility Reagent (Thermo Fisher Scientific). 10 l of cell lysates was incubated with 150 l of Pierce 660 nm Protein Assay Reagent for 5min, and the absorbance was measured at 660 nm using an iMark Microplate Reader (Bio-Rad Laboratories).”
Comment 2: The authors should include the detailed procedure for WB imaging and image quantification methods.
Response: We have added in the Methods: “The blots were placed in contact with FUJI Medical X-ray Film (FUJIFILM) in a cassette, and the light emitted during the chemiluminescent reaction exposed the film. The exposed films were then processed in a darkroom using developing solutions. To quantify the protein levels, the individual band images of proteins were scanned and analyzed using ImageJ software. Then, their intensities were normalized against FLAG or b-Actin as an internal control.” The band intensities of WB were indicated below each WB panel.
Comment 3: Please provide more details regarding the animals and animal number used in the study. There is no description of the total number of mice used for the study, N per experimental group, or N per analyzed endpoint in the Methods.
Response: Thank you for your thoughtful comment. We agree that the details of animals in the study are important, and have added in the Methods: “twenty mice were divided into four groups (n=5): control group, vincristine (VCR) group, VCR + low dose of carnosol group, and VCR + high dose of carnosol group. The sample size was determined based on previous studies to obtain a significance level. VCR was dosed intraperitoneally at 1 mg/kg in saline daily for 5 days. Carnosol was dosed orally by gavage at 10 and 100 mg/kg in vehicle (5% DMSO, 10% methyl-b-cyclodextrin) daily starting 2 h before VCR. The control group received only the solvent. On Day 8 after the start of administration, a von Frey mechanical threshold test, plasma collection and tissue collection were performed. In another study, RME-CA was used instead of carnosol, with all other conditions remaining the same.”
Comment 4: How was the sample size determined?
Response: The sample size was determined based on previous studies to obtain a significance level.
Comment 5: In immunostaining methods, the authors should include the details about quantification.
Response: We thank the reviewer for the comment, and have added in the Methods: For quantification of neurofilament integrity, the fluorescence intensities of NF-L, b3-tubulin and DAPI were scanned and analyzed using ImageJ software. Then, the fluorescence intensities of NF-L and b3-tubulin were normalized against nuclear numbers.” The quantification dates of immunostaining have shown in Fig 4 F, G and Fig S1 F, G.
Comment 6: In statistical analysis, the authors replaced the+/- with
Response: The +/- has replaced with ± symbol.
Comment 7: In Figure 1B, the authors should represent the loading control protein expression.
Response: Thanks for your valuable feedback. Because the two cell lines are different species, a CBB-stained gel of cell lysates was used as a loading control (Fig. 1B, right).
Comment 8: In Figure 3B and 4D, the WB images were from the same membrane? If not please represent individual loading control for each WB images.
Response: We thank the reviewer for the comment. The WB images were not from the same membrane. b-Actin has used as individual loading control, and the band intensities of the indicated proteins were normalized to their corresponding bands of b-Actin in the right panel, and the values were indicated below each WB panel.
Comment 9: In Figure 5E, the authors should include the merged images of PGP9.5 and DAPI.
Response: We have added the merged images of PGP9.5 and DAPI.
Comment 10: How the intraepidermal nerve fiber numbers was counted? Please include the details in materials and method section.
Response: We appreciate your feedback. We have added in the methods: “The density of the intraepidermal nerve fibers (IENF) was determined by counting the number of IENFs crossing the dermal-epidermal junction in three randomly selected slices per footpad. The length of the epidermis was measured using ImageJ software, and the numbers of the IENFs (fiber numbers/mm) was obtained.”
Round 2
Reviewer 1 Report
Authors have addressed all my comments and suggestions. This version can be considered to be pulbished.
Authors have highlighted all changes on the manuscript. They correspond all my comments.